# Overexpression of NOX2 Exacerbates AngII-Mediated Cardiac Dysfunction and Metabolic Remodelling

**DOI:** 10.3390/antiox11010143

**Published:** 2022-01-10

**Authors:** Synne S. Hansen, Tina M. Pedersen, Julie Marin, Neoma T. Boardman, Ajay M. Shah, Ellen Aasum, Anne D. Hafstad

**Affiliations:** 1Cardiovascular Research Group, Department of Medical Biology, Faculty of Health Science, UiT—The Arctic University of Norway, 9019 Tromsø, Norway; tipe@biomed.au.dk (T.M.P.); Julie.MARIN@student.umons.ac.be (J.M.); neoma.boardman@uit.no (N.T.B.); ellen.aasum@uit.no (E.A.); anne.hafstad@uit.no (A.D.H.); 2School of Cardiovascular Medicine & Sciences, King’s College London, British Heart Foundation Centre of Excellence, London SE5 9NU, UK; ajay.shah@kcl.ac.uk

**Keywords:** angiotensin II, NOX2, cardiac disease, hypertension, cardiac efficiency, cardiac hypertrophy

## Abstract

The present study aimed to examine the effects of low doses of angiotensin II (AngII) on cardiac function, myocardial substrate utilization, energetics, and mitochondrial function in C57Bl/6J mice and in a transgenic mouse model with cardiomyocyte specific upregulation of NOX2 (csNOX2 TG). Mice were treated with saline (sham), 50 or 400 ng/kg/min of AngII (AngII_50_ and AngII_400_) for two weeks. In vivo blood pressure and cardiac function were measured using plethysmography and echocardiography, respectively. Ex vivo cardiac function, mechanical efficiency, and myocardial substrate utilization were assessed in isolated perfused working hearts, and mitochondrial function was measured in left ventricular homogenates. AngII_50_ caused reduced mechanical efficiency despite having no effect on cardiac hypertrophy, function, or substrate utilization. AngII_400_ slightly increased systemic blood pressure and induced cardiac hypertrophy with no effect on cardiac function, efficiency, or substrate utilization. In csNOX2 TG mice, AngII_400_ induced cardiac hypertrophy and in vivo cardiac dysfunction. This was associated with a switch towards increased myocardial glucose oxidation and impaired mitochondrial oxygen consumption rates. Low doses of AngII may transiently impair cardiac efficiency, preceding the development of hypertrophy induced at higher doses. NOX2 overexpression exacerbates the AngII -induced pathology, with cardiac dysfunction and myocardial metabolic remodelling.

## 1. Introduction

Activation of the renin–angiotensin–aldosterone system (RAAS) is known to play an important role in a range of conditions known to increase the risk of developing cardiovascular diseases. Angiotensin (AngII) induces systemic effects, including arterial vasoconstriction as well as sodium and water retention, which may result in hypertension, increased cardiac workload, and development of heart failure. However, AngII has also been shown to have a direct effect on cardiomyocytes, affecting intracellular processes such as increased production of reactive oxygen species (ROS), fibrosis, hypertrophy, apoptosis, endoplasmic reticulum stress (ER stress), and inhibition of autophagy [1,2,3]. Therefore, AngII likely has a key role in cardiac remodelling and the development of cardiac dysfunction, also in the absence of hypertension.

In experimental studies of AngII-mediated heart failure, the AngII dose and treatment protocols are highly variable. High doses, >1000 ng/kg/min, typically cause an overt hypertension and hypertrophy [4,5,6], creating severe models of heart disease and possibly cachexia. Low doses, particularly those of ≤500 ng/kg/min, will cause an initial prehypertensive period characterized by auto potentiation to AngII, and subsequent development of hypertension, depending on the duration of treatment [7,8]. Although the phenotypic changes are easily studied in models using pressor doses, lower doses (no or slow pressor doses) are required to study the initial effect of AngII, without introducing confounding disease factors that may mask the direct effects of AngII. Additionally, although the suppression of the RAAS is an important medical therapy in heart failure patients, this alone does not prevent the progression, even in optimally treated patients. Thus, investigating the development of early AngII-induced disease can be an important step to discover relevant treatment options and novel therapies.

Altered substrate metabolism and loss of metabolic flexibility are hallmarks in heart failure [9,10,11,12,13,14]. In addition, decreased cardiac efficiency is an early indicator of the failing heart, often preceding development of cardiac dysfunction [12,14]. Although it is well known that AngII induces structural remodelling in the heart [15,16], its effect on cardiac metabolic remodelling is less clear. There is evidence showing that the prevention of metabolic alterations in the failing heart is beneficial [17]; thus, studying the metabolic adaptations and possible maladaptation in early AngII-induced disease could provide important insight into the therapeutic potential. A common mediator of AngII-induced processes is increased production of ROS, which is detectible even with very low doses of AngII and in early stages of heart failure [3,7,18,19]. Although there are several sources of ROS in cardiomyocytes, studies have confirmed that NADPH oxidase 2 (NOX2) is a crucial contributor to AngII-induced ROS production in the pathogenesis of heart failure [20,21,22]. We have previously found elevated myocardial ROS levels to be associated with increased myocardial oxygen consumption [14], and abrogation of NOX2 was shown to reduce cardiac ROS levels and improve cardiac efficiency in obesity-induced heart-failure [23]. Furthermore, non-pressor doses of AngII (50 ng/kg/min) have been shown to induce mitochondrial uncoupling in skeletal muscles [19], which could suggest an impact on mitochondrial efficiency.

The aim of the present study was therefore to examine the effects of a non-pressor dose of AngII (50 ng/kg/min) as well as a slow pressor dose of AngII (400 ng/kg/min) on cardiac function, substrate utilization, efficiency, and mitochondrial respiration. We also included mice with cardiomyocyte specific NOX2 overexpression to investigate whether NOX2 exacerbates AngII-induced cardiac metabolic remodelling.

## 2. Materials and Methods

### 2.1. Animal Models

Male 11-week-old C57BL/6J mice (Charles River Laboratories, Sulzfeld, Germany) were used in the study. In addition, age-matched male wild-type (WT) and transgenic mice with a cardiomyocyte-specific overexpression of NOX2 (csNOX2 TG), obtained from Professor Ajay M. Shah’s lab (King’s College, London, UK) were also included. The csNOX2 were created on a C57BL/6J background by cloning a 1.8 kb human NOX2 cDNA downstream of the myosin light chain-2 promoter, prior to injection into fertilized oocytes [22]. The expression of the NOX2 protein is approximately five times higher in csNOX2 TG mice than in WT mice, but the basal NOX2 activity and cardiac phenotype are unaltered. However, in response to pathological stressors, such as AngII stimulation, cardiomyocytes from csNOX2 TG mice exhibit increased NOX2-mediated ROS production [22].

All mice were acclimatized to the animal facilities for one week and kept on a 12:12 hour reversed light–dark cycle, in a room with a constant temperature of 21 °C and 55% humidity. The animals were given ad libitum access to a normal chow diet and water and were otherwise treated in accordance with the guidelines on accommodation and care of animals given by the European Convention for the Protection of Vertebrate Animals for Experimental and Other Scientific Purposes.

C57BL/6J WT and csNOX2 TG mice were given either saline (sham), non-pressor dose, 50 ng/kg/min AngII (AngII_50_) or slow pressor dose, 400 ng/kg/min AngII (AngII_400_). Micro-osmotic pumps (Model 1002, Alzet, Cupertino, CA, USA) were inserted subcutaneously, and the treatment lasted for two weeks. Animals were given a standard volume of 100 µL of saline or AngII (A9525, Sigma Aldrich, Saint-Louis, MO, USA). Prior to pump implantation, animals were given buprenorphine (0.05 mg/kg SC) as an analgesic. Experiments were done in our laboratory at the UiT—The Arctic University of Norway and were approved by the Animal Welfare Committee at the university and the Norwegian Food Safety Authority (FOTS id: 7435).

### 2.2. Echocardiography and Blood Pressure Measurements

Echocardiographic measurements were performed at baseline and repeated after two weeks. Animals were lightly anaesthetized with isoflurane (1.5–2% isoflurane) while lying in a supine position on a heated platform [24]. Measurements were obtained and analysed from parasternal short-axis M-mode and, for a more comprehensive examination, apical four-chamber Doppler images, as previously described [24]. A blinded operator imaged all mice and performed the subsequent analyses.

Blood pressures were obtained using tail-cuff plethysmography (Coda High Throughput Non-Invasive Blood Pressure System, Kent Scientific, Torrington, CT, USA) on awake animals.

### 2.3. Isolated Heart Perfusions

Isolated heart perfusion was performed the day after the last echocardiography. Animals were anesthetized with pentobarbital (100 mg/kg i.p.) and heparin (100 U, i.p.), and hearts were excised and placed in ice-cold buffer before being fixed in a perfusion system, in working heart mode. A modified Krebs–Henseleit buffer containing 5 mM of glucose and 0.4 mM of palmitate bound to 3% fat free bovine serum albumin (EQBAH66 Europa Bioproducts, Cambridge, UK) was used with ^4^C-U-labelled glucose (NEC04B005MC, Perken Elmer, Boston, MA, USA) and ^3^H-9,10-labelled palmitate (NET043005MC, Perken Elmer, Boston, MA, USA) in order to measure fatty acid and glucose oxidation rates [13]. Preload and afterload pressures were kept at a constant standardized level throughout the protocol (10 mmHg and 55 mmHg, respectively). A pressure catheter (Codman Microsensor, DePuy Synthes Co, MA, USA) was placed in the aortic line close to the heart, measuring peak systolic pressure (PSP). Left ventricular (LV) stroke work was calculated as the product of stroke volume * (PSP-filling pressure). Data were obtained and analysed using LabChart 7Pro software (AD Instruments, Bella Vista, Australia).

Fibre-optic O_2_ sensors (FOXY-AL 300; Ocean Optics, Duiven, The Netherlands) were placed in the buffer flow above the aorta (i.e., in the buffer entering the coronary vessels) and in the pulmonary trunk (in the buffer leaving the heart), to obtain the arterial–venous difference in PO_2_, in order to measure myocardial oxygen consumption (MVO_2_) as previously described [25]. Total mechanical efficiency was calculated as the ratio between LV stroke work and MVO_2_ [14]. Finally, MVO_2_ was also measured in unloaded retrograde perfused hearts (MVO_2Unloaded_) and in electrically arrested hearts to measure MVO_2_ for basal metabolism (MVO_2BM_). MVO_2_ for processes associated with excitation–contraction coupling (MVO_2ECC_) was calculated as MVO_2unloaded_-MVO_2BM_ [26]. After ex vivo perfusion protocol, heart tissue samples were harvested for mRNA analysis and respirometry measurements.

### 2.4. Real-Time Quantitative PCR 

LV tissue from perfused hearts was immersed in RNAlater (Qiagen, Hilden, Germany), and total RNA was extracted according to the RNeasy Fibrous Tissue kit Protocol (Qiagen Nordic, Oslo, Norway). Real-time qPCR analysis was performed on tissue samples using an ABI PRISM 7900 HT Fast real-time thermal cycler as previously described [27]. Details about primer/probe sequences are given in Appendix A.

### 2.5. Respirometry in Frozen Samples

Oxygen consumption rates (OCR) were measured in homogenates from frozen (−70 °C) LV biopsies according to the method previously described by Acin-Perez et al., 2020 [28]. This method has been shown to allow assessment of OCR in the respiratory chain complexes comparable to uncoupled mitochondria from fresh tissue [28]. We added homogenate to the closed chambers of the oxygraph (O2K, Oroboros Instrument, Innsbruck, Austria). Data were recorded using DatLab 5 software (Oroboros Instrument, Innsbruck, Austria). Measurements were done at 37 °C. Two respiration protocols were performed in two separate chambers after recording a stable basal respiration. Protocol A: 1 mM NADH (Sigma Aldrich, Saint-Louis, MO, USA) was added to the chamber to stimulate complex I (CI) respiration. Then, 0.5 µM rotenone (Sigma Aldrich, Saint-Louis, MO, USA) was added to inhibit CI respiration. Protocol B: 0.5 µM rotenone was added to the oxygraph chamber, followed by 10 mM succinate to assess complex II (CII) respiration. Finally, 5 mM malonic acid (Sigma Aldrich, Saint-Louis, MO, USA) and 2.5 mM antimycin A (Sigma Aldrich, Saint-Louis, MO, USA) were added sequentially to inhibit CII and complex III (CIII), respectively, and to assess non-mitochondrial residual OCR (ROX). The recorded O_2_ flux in each state was normalized to protein concentration quantified by Bradford protein Assay (Bio-Rad Laboratories, Hercules, CA, USA).

### 2.6. Blood Glucose Levels

Blood was drawn after two weeks of treatment at the point of euthanasia. We used a standard blood glucose measuring device (Freestyle, Blood glucose measuring system, Abbott Park, IL, USA).

### 2.7. Statistics

The results are presented as mean ± standard error of means in tables, line charts, and column bar graphs. When comparing differences between groups or individuals, unpaired and paired Student’s t-tests were performed, respectively.

## 3. Results

Two weeks of AngII treatment (50 or 400 ng/kg/min) did not alter body weight, blood glucose, or liver weight in C57BL/6J mice. Following AngII_50_ treatment there were no signs of cardiac hypertrophy; however, using the slow pressor dose (AngII_400_), hypertrophy was evident as increased heart weight and increased cardiac mRNA expression of the gene encoding for the hypertrophic markers natriuretic peptide A(*nppa*) and natriuretic peptide B (*nppb*) (Table 1). We did not find AngII_400_ to alter the mean arterial pressure (MAP), but there was a modest elevation of the arterial systolic blood pressure (SBP) at two weeks of treatment when compared to baseline (106 ± 3 vs. 94 ± 4 mmHg, respectively, *p* < 0.05).

Cardiomyocyte specific upregulation of NOX2 did not aggravate the effects of Ang_400_ treatment in terms of body weight, cardiac hypertrophy, or expression of hypertrophic markers in csNOX2 TG mice (Table 1). Both the SBP and the MAP were slightly elevated compared to baseline in AngII_400_ treated csNOX2 TG mice (SBP 110 ± 4 vs. 90 ± 2 mmHg, and MAP 90 ± 4 vs. 72 ± 2 mmHg, *p* < 0.05) (Table 1).

Baseline measurements of in vivo cardiac parameters were not different between treatment groups or genetic phenotype (Table 2 and Table 3). Echocardiographic measurements of cardiac dimensions support the findings of AngII_400_-induced hypertrophy with increased LV posterior wall thickness in diastole (LVPW;d) and increased LV mass (Table 2). The AngII_400_ induced hypertrophy was not associated with deterioration of cardiac function in C57BL/6J mice.

Again, overexpression of csNOX2 did not aggravate the hypertrophic effects of AngII_400_, as AngII_400_ LV mass and posterior wall thickness were not altered in the csNOX2 TG mice as compared to WT mice. However, the ratio between end-diastolic volume and LV mass was elevated in csNOX2 TG AngII_400_ mice, signifying an eccentric hypertrophy with dilation and increased intraventricular volume. Upregulation of csNOX2 deteriorated LV systolic function, as it increased the systolic LV internal diameter in csNOX2 TG AngII_400_, but not WT AngII_400_ mice. This led to a decrease in fractional shortening (FS) in csNOX2 TG AngII_400_ mice (Table 3). Similarly, csNOX2 TG AngII_400_ mice also showed an increase in end-systolic volume (ESV), resulting in a decreased ejection fraction (EF) (Table 3). In addition, AngII_400_ was found to significantly increase the E/E’ ratio only in csNOX2 TG. As elevation of the E/E’ ratio indicates increased LV filling pressure and decreased compliance, these data suggest progression of LV diastolic dysfunction. Thus, although hypertrophy was not exacerbated per se, the pathological phenotype was aggravated, signifying progression of cardiac disease in the csNOX2 TG mice. Interestingly, ex vivo function measured in isolated working hearts did not reveal differences in ventricular function between treatment groups or genotypes (Appendix A).

In this study, we also investigated the effects of AngII-treatment on myocardial energetics. We found a significant reduction in cardiac mechanical efficiency in AngII_50_ mice compared to sham mainly due to increased myocardial oxygen consumption in unloaded hearts (MVO_2unloaded_, *p* = 0.085) (Figure 1A,D). Further, increasing the dose of AngII did not impair myocardial energetics as there were no differences in mechanical efficiency, MVO_2unloaded_, or MVO_2_ for processes associated with excitation–contraction coupling (MVO_2ECC_) or basal metabolism (MVO_2BM_). Furthermore, upregulation of NOX2 did not alter the response to AngII_400_ with regards to mechanical efficiency or MVO_2_.

Altered myocardial substrate utilization has previously been linked to myocardial energetics [11,29], and we therefore assessed myocardial glucose and fatty acid oxidation rates in response to AngII treatment. Neither AngII_50_ nor AngII_400_ was found to alter substrate oxidation rates (Figure 2). This was supported by unchanged gene expression of markers of metabolic reprogramming, such as peroxisome proliferator-activated receptor α (*pparα*), *cd36*, protein pyruvate dehydrogenase kinase 4 (*pdk4*), and hexokinase (*hk*) (Appendix A). Cardiac overexpression of NOX2, however, induced a metabolic shift in response to AngII treatment, as we found increased glucose oxidation rates in csNOX2 TG AngII_400_ (Figure 2F). This metabolic reprogramming was associated with a significant increase in *pdk4* mRNA levels in csNOX2 TG. There were no differences in other markers of metabolic reprogramming (*pparα*, *cd36*, *ldh* and *hk*) between AngII_400_ treated csNOX2 TG and WT (Appendix A).

The oxygen consumption rate (OCR) through the complexes in the electron transport system in the mitochondria was measured in homogenates from LV heart tissue that had been previously frozen. There was no difference in OCR in homogenates from sham and AngII-treated LV heart tissue (Figure 3A,B). In the csNOX2 TG, AngII_400_ resulted in a significant reduction in basal and residual oxygen consumption rates (ROX) compared to WT (Figure 3C). In addition, there was a non-significant tendency towards lower OCR in complex I (CI) in homogenate from csNOX2 TG AngII_400_, which might indicate altered mitochondrial respiration.

## 4. Discussion

While it is well documented that high doses of AngII lead to a rapid and overt pressor response accompanied by development of cardiac dysfunction [4,5,6], the functional effect of low doses and their impact on cardiac metabolism has been less described. In the present study we have examined metabolic and functional changes in the heart associated with two-week treatment of mice using non-pressor dose (50 ng/kg/min) or slow pressor dose (400 ng/kg/min) of AngII. These doses were used to mimic early phase of heart failure, prior to the introduction of confounding effects that follow the complexity of overt hypertension. Knowing that NOX2 is an important target for AngII in the heart, we also included a transgenic model with cardiomyocyte specific NOX2 overexpression (csNOX2 TG) to evaluate the potential role of NOX2.

We did not detect changes in body weight development following AngII_50_ or AngII_400_ treatment, similar to other reports [19]. These data support that there were no cachexic effects of the doses used as compared to the effect of higher doses [30]. Previously, studies using slow pressor doses (400 and 500 ng/kg/min) have reported elevations in systolic blood pressure (SBP) and mean arterial pressure (MAP) [7,31,32]. We did not detect changes in MAP but did observe a slight increase in SBP, which was subtle compared to previous reports [7,31,32]. It should be noted, however, that there are limitations in sensitivity when using tail-cuff plethysmography compared to more invasive measurements, and this could explain the inconsistencies in reports regarding the effect of slow pressor dose of AngII on blood pressure.

In a study by Byrne and colleagues, 2003 [21], AngII-mediated elevation of blood pressure was linked to the expression of NOX2, as mice with a global knockdown of NOX2 (gp91^phox-/-^ mice) did not exhibit the same increased SBP in response to AngII-pressor doses. A previous study on csNOX2 TG mice treated with 300 ng/kg/min of AngII did not report any blood pressure differences between csNOX2 TG and WT mice [22]. In the present study, both MAP and SBP were slightly increased in the csNOX2 TG mice, which could be linked to increased cardiac output during the awake blood pressure measurements.

AngII_50_ did not induce any hypertrophic changes in the myocardium, in line with previous reports [19]. Mice treated with AngII_400_, however, display a hypertrophic phenotype, confirmed by increased wall thickness and increased gene expression of *nppa* and *nppb*. Our data therefore show that AngII can mediate direct hypertrophic effects also in the absence of overt changes in MAP. This also supports other studies on using low doses of AngII (50–300 ng/kg/min) for two to four weeks where mice developed cardiac hypertrophy and signs of fibrosis, despite an absence of increased blood pressure [6,20,22,33].

Slow pressure doses of AngII have been shown to increase both ROS production and NOX2 activity during the development of cardiac hypertrophy [31]. In addition, Byrne et al., 2003 [21] demonstrated that NOX2 is essential for the development of AngII-induced hypertrophy, as NOX2 KO mice failed to develop hypertrophy even in the presence of pressor doses of AngII (>1000 ng/kg/min). Additionally, Zhang et al., 2015 [22], using slightly lower doses of AngII (300 ng/kg/min for two weeks), reported aggravated hypertrophic response in csNOX2 TG mice. In contrast to this, we could not detect any differences in hypertrophy in csNOX2 TG or WT mice in the present study.

Non-pressor doses (50 ng/kg/min) of AngII did not cause changes in LV functional parameters, in coherence with the findings of Inoue and colleagues [19]. However, reduced diastolic function, both with and without systolic dysfunction, has previously been reported following treatment with both 150 ng/kg/min [6] and 500 ng/kg/min [32] of AngII. This contrasts with the current study, where we could not detect any in vivo or ex vivo ventricular dysfunction in AngII_400_ treated mice. We did however observe that an overexpression of NOX2 lead to AngII-mediated cardiac dysfunction as indicated by increased end-systolic volume (ESV) accompanied by reduced ejection fraction (EF) in csNOX2 TG mice. In addition to the systolic dysfunction, these hearts also showed increased LV filling pressure and reduced compliance, indicative of diastolic dysfunction. There are several plausible mechanisms behind the development of LV dysfunction, as NOX2-induced ROS production has been shown to impair calcium handling [22,34,35] and induce a range of pathological cardiac changes such as fibrosis, apoptosis, and hypertrophy [36,37]. Our findings are consistent with those of Zhang et al. [22], who showed that prolonged activation of NOX2 in csNOX2 TG mice resulted in deterioration of cardiac function similar to the current results. Taking these studies together, our data support the notion that the functional consequence of increased NOX2 activation in the heart may depend on type and proportion of stress applied to the heart [38].

Pathological hypertrophy and heart failure are known to be associated with changes in myocardial substrate utilization [39]. We did not find any changes in oxygen consumption rates (OCR) in ventricular homogenates or a shift in myocardial substrate oxidation rates in non-transgenic mice treated with AngII_50_ or AngII_400_. Although changes in myocardial substrate utilization have been reported following chronic exposure to AngII, there are discrepancies in terms of substrate preference [16,17,40,41]. A study utilizing slightly higher doses of AngII (<800 ng/kg/min for 2–4 weeks) reported reduced in myocardial fatty acid oxidation rates with increased utilization of glucose [17]. Moreover, studies using pressor doses (>1000 ng/kg/min for two weeks) generally report systemic insulin resistance, with a subsequent increased cardiac preference for fatty acids and diminished glucose and lactate oxidation [16,40]. Although the metabolic phenotype in these studies varies, the AngII-treated mice all displayed ventricular dysfunction [16,17,40,41], suggesting that a metabolic shift becomes evident at the onset of cardiac failure. The present study shows that C57Bl/6J mice treated with the AngII_400,_ showed neither cardiac dysfunction nor changes in myocardial substrate utilization, further supporting the notion that the metabolic alterations occur only when the AngII-induced pathological condition has progressed far enough to manifest as cardiac dysfunction.

In the csNOX2 TG, AngII_400_ treatment led to a significant reduction in basal OCR in ventricular homogenates. Previous reports have shown that AngII-induced oxidative stress results in mitochondrial damage and dysfunction [3]. In addition, we found an increase in myocardial glucose oxidation, accompanied by a tendency towards reduced palmitate oxidation in ex vivo csNOX2 TG hearts. There was also a significant increase in *pdk4*, a marker of metabolic switch. To our knowledge, no one has previously investigated the metabolic changes in response to slow pressor dose of AngII in csNOX2 TG mice, but our finding corroborates the metabolic shift previously reported in animal models with AngII-induced heart failure [17,41]. Oxidative stress has previously been associated with induction of translocation of GLUT4 to the plasma membrane, deacetylation of the pyruvate dehydrogenase complex, and enhanced glucose utilization [40,42,43,44]. Interestingly, Pellieux and collaborators reported that AngII-treatment led to downregulation of several key regulatory proteins of fatty acid oxidation and that these effects were impeded by inhibition of ROS production [45]. Accordingly, the overexpression and subsequent increase in ROS production following AngII treatment in the csNOX2 TG animals in this study could mediate downregulation of regulatory proteins and cause the observed substrate shift through the same signalling pathways that were described by Pellieux and Aikawa with colleagues [42,45].

In a previous study performed by our group, we demonstrated that ablation and pharmacological inhibition of NOX2 in obese mice improved the mechanical efficiency and reduced myocardial oxygen consumption (MVO_2_) for non-mechanical cardiac work [23]. We have also previously demonstrated an association between increased ROS and increased MVO_2_ [14], which together suggest a link between myocardial oxygen wasting and NOX2 activation. Increased NOX2 activity has in several studies been linked to altered calcium handling [22,34,35] such as increased calcium leak through the Ryanodine Receptor (RyR) in the sarcoplasmic reticulum [34,35]. This could potentially lead to oxygen wasting processes in the myocardium. Thus, we were surprised to find that the slow pressor dose, AngII_400_, did not cause altered mechanical efficiency or altered MVO_2_ in neither non-transgenic nor in the csNOX2 TG mice. In contrast, we observed a decline in mechanical efficiency in the AngII_50_ mice. In addition, these hearts showed a somewhat higher unloaded MVO_2_. As they did not show altered oxygen cost of excitation contraction coupling (ECC), it suggests that the decline in mechanical efficiency most likely is caused by a higher basal metabolism and/or a higher work-dependent oxygen cost. The same treatment has previously showed increased oxygen wasting processes in skeletal muscle in mice, including increased residual oxygen consumption (ROX) as well as increased expression of uncoupling proteins in mitochondria [19]. Although there was a tendency for increased OCR and ROX in the AngII_50_ group, ROX was unaltered in AngII_400_ and reduced in csNOX2 TG AngII_400_, suggesting that there might indeed be transient effects of AngII-mediated signalling in the myocardium.

The current study only includes male mice, and therefore we acknowledge the gender bias. It is well known that there are differences between males and females in symptoms and frequency of heart failure [46]. In addition, sex steroids have been shown to influence the progression of AngII-mediated cardiovascular diseases [31,47]. In specific, AngII only led to increased NOX2 activity in cardiac tissue from male and ovariectomized female mice as compared to control female mice [31], suggesting oestrogens to protect against NOX2 upregulation. Therefore, more studies on gender differences in cardiometabolic adaptations to cardiac stressors are warranted in the future.

## 5. Conclusions

This study shows that cardiac efficiency may be reduced by non-pressure doses of AngII (50 ng/kg/min), preceding apparent cardiac hypertrophy, ventricular dysfunction, or changes in myocardial substrate utilization. Interestingly, a slow pressor dose of AngII (400 ng/kg/min), which did induce cardiac hypertrophy, was not associated with impaired cardiac efficiency. This dose did not impact ventricular function or myocardial substrate utilization in WT mice. However, the same slow pressure dose, in mice with cardiomyocyte-specific overexpression of NOX2, led to cardiac dysfunction and metabolic reprogramming without any apparent effect on cardiac energetics. Our data therefore suggest that impaired cardiac energetics may precede AngII-induced ventricular structural and metabolic remodelling. Hence, increased NOX2 activity may aggravate metabolic as well as structural cardiac remodelling of AngII-mediated signalling.

## Figures and Tables

**Figure 1 antioxidants-11-00143-f001:**
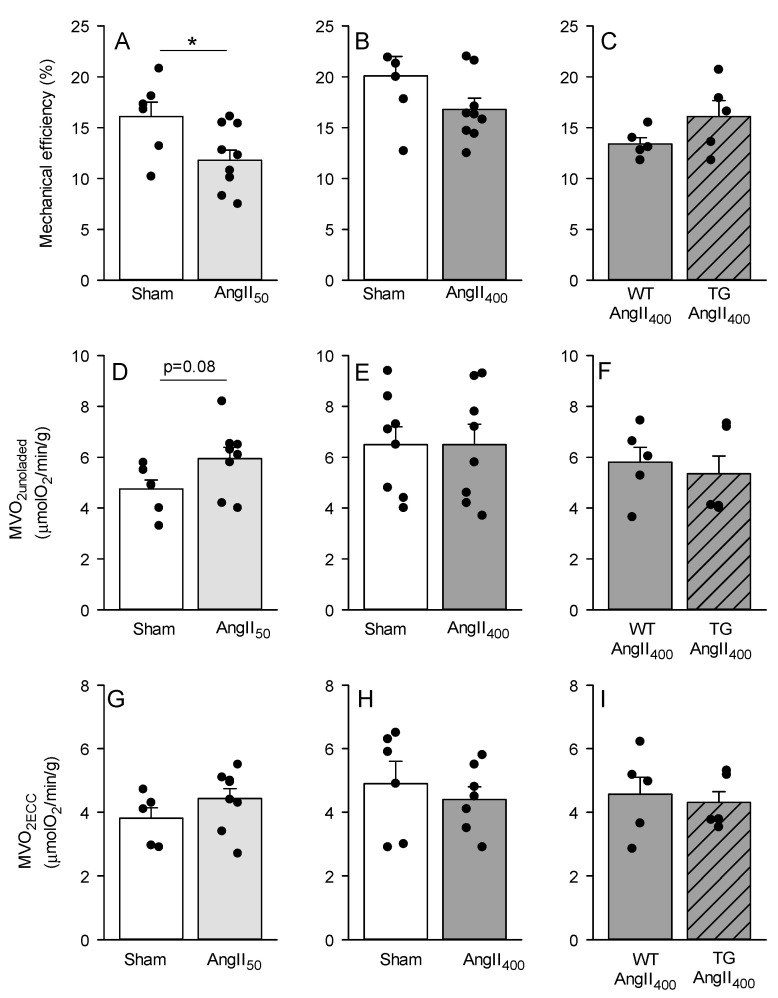
Mechanical efficiency (**A**–**C**), myocardial oxygen consumption in mechanically unloaded hearts (MVO_2unloaded_, **D**–**F**) and MVO_2_ for processes associated with excitation-contraction coupling (MVO_2ECC_, **G**–**I**), measured in isolated perfused hearts from C57BL/6J, wild-type (WT), and csNOX2 transgenic (TG) mice treated for two weeks with micro-osmotic pumps containing either saline (sham), 50 or 400 ng/kg/min angiotensin II (AngII_50_ and AngII_400_). The data are presented as mean ± SEM. * *p* < 0.05 vs. sham.

**Figure 2 antioxidants-11-00143-f002:**
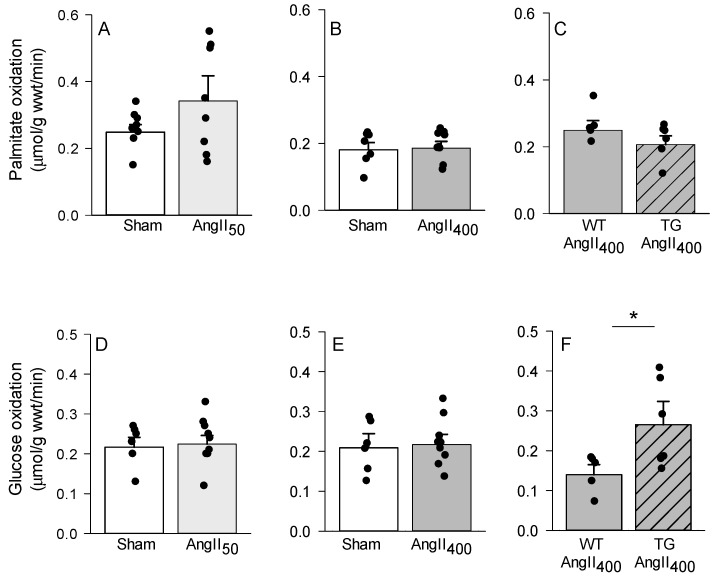
Palmitate (**A**–**C**) and glucose (**D**–**F**) oxidation rates assessed in isolated working hearts from C57BL/6J, wild-type (WT), and csNOX2 transgenic (TG) mice treated for two weeks with micro-osmotic pumps containing either saline (sham), 50 or 400 ng/kg/min angiotensin II (AngII_50_ and AngII_400_). The data are presented as mean ± SEM. * *p* < 0.05 vs. WT.

**Figure 3 antioxidants-11-00143-f003:**
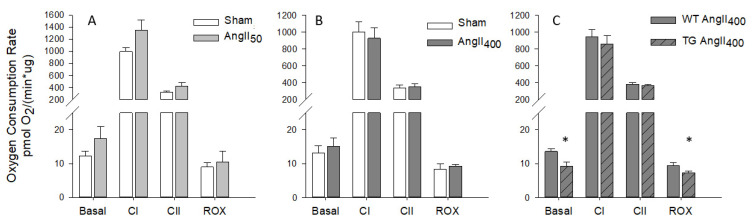
Oxygen consumption rate measured in homogenate from frozen left ventricular heart tissue from C57BL/6J, wild-type (WT), and cardiomyocyte specific NOX2 transgenic (TG) mice treated for two weeks with micro-osmotic pumps containing either saline (sham), 50 or 400 ng/kg/min angiotensin II (AngII_50_ and AngII_400_). (**A**) Sham and AngII_50_. (**B**) Sham and AngII_400_. (**C**) WT AngII_400_ and TG AngII_400_. Basal, homogenate with cytochrome C; Complex I (CI), homogenate and NADH; Complex II (CII) is homogenate with rotenone (CI-blocker) and succinate; Residual Oxygen consumption (ROX), homogenate with malonate (CII-blocker) and antimycin A (CIII-blocker). Data are means ± SEM. * *p* < 0.05 vs. WT.

**Table 1 antioxidants-11-00143-t001:** Animal characteristics of C57BL/6J, wild-type (WT), and csNOX2 transgenic (TG) mice treated for two weeks with micro-osmotic pumps containing either saline (sham), 50 or 400 ng/kg/min angiotensin II (AngII_50_ and AngII_400_). The data are presented as mean ± SEM.

	Sham	AngII_50_	Sham	AngII_400_	WT AngII_400_	TG AngII_400_
*n*	9	11	10	10	6	6
MAP (mmHg)	n.m.	n.m.	99 ± 6	105 ± 6	86 ± 3	90 ± 4 ^#^
SBP	n.m.	n.m.	136 ± 12	131 ± 7	107 ± 3 ^#^	110 ± 4 ^#^
Body weight (g)	26 ± 0.3	27 ± 0.2	25 ± 0.3	25 ± 0.4	27.2 ± 0.8	27.0 ± 0.2
Liver weight (g)	1.33 ± 0.7	1.48 ± 0.05	0.99 ± 0.03	1.00 ± 0.04	0.95 ± 0.07	0.96 ± 0.04
Blood glucose (mM)	5.2 ± 0.3	4.6 ± 0.3	5.8 ± 0.5	5.5 ± 0.3	6.2 ± 0.6	6.2 ± 0.5
HW/BW (mg)	5.1 ± 0.2	4.9 ± 0.1	5.1 ± 0.1	5.5 ± 0.1 *	6.0 ± 0.2	5.7 ± 0.3
*nppa_heart_*	1.0 ± 0.1	1.2 ± 0.2	1.0 ± 0.2	2.5 ± 0.3 *	1.0 ± 0.2	0.7 ± 0.2
*nppb_heart_*	1.0 ± 0.1	1.3 ± 0.2	1.0 ± 0.1	1.4 ± 0.1 *	1.0 ± 0.1	0.9 ± 0.1

Blood samples were obtained from fed animals. The cardiac tissue mRNA expression of genes encoding for Natriuretic Peptide A (*nppa_heart_*) and Natriuretic Peptide B (*nppb_heart_*) were normalized to the corresponding expression in respective sham C57Bl-6J or WT AngII_400_; heart weight/body weight, HW/BW; MAP, mean arterial pressure. * *p* < 0.05 vs. sham, ^#^
*p* < 0.05 vs. baseline.

**Table 2 antioxidants-11-00143-t002:** In vivo left ventricular function assessed by transthoracic echocardiography in C57BL/6J mice treated with slow pressure dose of angiotensin II (AngII_400_) or saline (sham) for two weeks. Measurements were obtained and analysed from parasternal short-axis M-mode. Data are presented as mean ± SEM.

	Sham		AngII_400_	
	Baseline	Week 2	Baseline	Week 2
*n*	7	7	7	7
Heart rate (BPM)	460 ± 21	451 ± 13	470 ± 9	458 ± 19
LVPW;d (mm)	0.71 ± 0.04	0.74 ± 0.02	0.77 ± 0.02	0.81 ± 0.02 *
LVID;d (mm)	3.7 ± 0.1	3.9 ± 0.1	3.7 ± 0.1	4.0 ± 0.1 ^#^
LV mass (mg)	74 ± 7	88 ± 3 *	82 ± 2	102 ± 3 ^#,^*
LV Mass/BW (mg/g)	3.1 ± 0.2	3.9 ± 0.1	3.6 ± 0.1	4.4 ± 0.2
LVEDV (μL)	57 ± 3	65 ± 4	59 ± 3	69 ± 2 ^#^
LVESV (μL)	19 ± 1	22 ± 3	20 ± 2	24 ± 2
SV (μL)	38 ± 2	43 ± 2 ^#^	39 ± 2	45 ± 2 ^#^
FS (%)	37 ± 1	36 ± 1	36 ± 1	36 ± 2
EF (%)	67 ± 1	67 ± 2	66 ± 2	66 ± 3
LV Volume/LV Mass (µL/mg)	0.78 ± 0.05	0.74 ± 0.04	0.69 ± 0.04	0.68 ± 0.01

LVPW;d, left ventricular (LV) posterior wall thickness; LVID;d, LV internal diameter in diastole; BW, body weight; EDV and ESV, end-diastolic and end-systolic volumes; SV, stroke volume; FS, fractional shortening; EF, ejection fraction. ^#^
*p* < 0.05 vs. baseline, * *p* < 0.05 vs. sham.

**Table 3 antioxidants-11-00143-t003:** In vivo left ventricular function assessed by transthoracic echocardiography in wild-type (WT) and csNOX2 transgenic (TG) mice treated with angiotensin II (AngII_400_) for two weeks. Measurements were obtained and analysed from parasternal short-axis M-mode and apical four-chamber view. Data are presented as mean ± SEM.

	WT AngII_400_		TG AngII_400_	
	Baseline	Week 2	Baseline	Week 2
*n*	4	6	6	5
Heart rate (BPM)	491 ± 12	526 ± 14	484 ± 7	515 ± 13 ^#^
LVPW;d (mm)	0.82 ± 0.05	0.92 ± 0.05 ^#^	0.77 ± 0.02	0.89 ± 0.02 ^#^
LVID;d (mm)	4.1 ± 0.1	4.0 ± 0.1	4.2 ± 0.1	4.3 ± 0.2
LV mass (mg)	101 ± 5	118 ± 4 ^#^	95 ± 4	113 ± 6 ^#^
LV Mass/BW (mg/g)	3.6 ± 0.1	4.3 ± 0.1 ^#^	3.5 ± 0.1	4.2 ± 0.1 ^#^
LVEDV (μL)	73 ± 3	71 ± 6	77 ± 2	84 ± 8
LVESV (μL)	27 ± 1	30 ± 4	32 ± 2	43 ± 5 ^#,^*
SV (μL)	45 ± 2	40 ± 2	45 ± 2	41 ± 3
FS (%)	33 ± 1	30 ± 2	31 ± 1	25 ± 1 ^#,^*
EF (%)	62 ± 2	58 ± 2	59 ± 2	50 ± 2 ^#,^*
LV Volume/LV Mass (µL/mg)	0.72 ± 0.04	0.60 ± 0.05	0.83 ± 0.02 *	0.74 ± 0.04 ^#,^*
E/A	1.3 ± 0.1	1.5 ± 0.2	1.4 ± 0.0	1.4 ± 0.2
E/E’	28 ± 2	30 ± 2	29 ± 1	34 ± 2 ^#^
Deceleration time (ms)	24 ± 1	20 ± 2	23 ± 2	19 ± 2

LVPW;d, left ventricular (LV) posterior wall thickness; LVID;d, left ventricular (LV) internal diameter in diastole; EDV and ESV, end-diastolic and end-systolic volumes; SV, stroke volume; FS, fractional shortening; EF, ejection fraction; E/A, ratio of velocity of early to late ventricular filling; E/E’; ratio of velocity of early ventricular filling to early diastolic mitral annular velocity. ^#^
*p* < 0.05 vs. baseline, * *p* < 0.05 vs. WT.

## Data Availability

The data is contained within the article or Appendix A.

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
