# Peer review of "Overexpression of NOX2 Exacerbates AngII-Mediated Cardiac Dysfunction and Metabolic Remodelling"

_antioxidants, 2022, doi:10.3390/antiox11010143_

Round 1

Reviewer 1 Report

Title: ‘Overexpression of NOX2 exacerbates AngII-mediated cardiac dysfunction and metabolic remodeling.’

 Hansen et al. used micro-osmotic pumps to introduction low doses of AngII in C57BL/6 WT and cardiac specific overexpressed NOX2 transgenic (csNOX2TG) mice at the rate of 50ng/kg/min (AngII50) or 400ng/kg/min (AngII400) for two weeks. The authors found that, in WT mice, AngII400 was able induced cardiac hypertrophy marked by increased heart weight/body weight ratio and increase mRNA expression of hypertrophic markers (nppa and nppb) without significant elevation in blood pressure. There was little or no effect on cardiac function, efficiency, or substrate utilization. However, in csNOX2TG, AngII400 induced cardiac dysfunction, not observed in WT mice, leading to decreased fatty acid oxidation, increase glucose oxidation and impaired mitochondrial oxygen consumption rates. It is an interesting study, but the authors fail to show the mechanism by which NOX2 overexpression exacerbates the effects of AngII400.

Major comments:

  • Why only male mice? Explain the rationale. Also, include a limitation section.
  • One of the mouse models used was a cardiac specific overexpression NOX2 model. So, it left to wonder why cardiomyocytes was not used in this study. It would be interesting to assess the effects of NOX2 overexpression at the cellular level. Would there be an increase or decrease in contractility in the presence of low dose of Ang II? Measurement of calcium transient amplitude.
  • What is the mechanism by which NOX2 overexpression, in the presence of Ang II, causes an increase in LVESV resulting in decreased FS? Is it through increase total cell ROS or mitochondrial ROS? Increased mitochondrial calcium?
  • What role does the stress kinases play in this pathway?
  • How are the major ion channels (Ryanodine receptor, SERCA and LTCC) affected?
  • It is a cardiac specific model. Why use ventricular homogenates which have multiple cell types? It would have been preferable to used isolated cardiomyocytes from AngII treated mice. Explain the rationale.
  • In Table 1, values for SBP were not found and the values for MAP were different from what was stated in the text. How do you explain the discrepancies?
  • Pdk4 was significantly increased at the mRNA level in csNOX2TG compared to WT after Ang II treatment. However, there is not mention of the expression of level of PDK4 protein levels. Is PDK4 affected at the protein level?
  • There is a trend towards a decreased in palmitate oxidation in csNOX2TG. It would be interesting to examine the effects on proteins involved in fatty acid oxidation.

        Minor comments:
    Typos- Materials and Methods: cxNOX2 should be csNOX2; Discussion: SPB should be SBP and Ang400 to AngII400.
  • In Materials and Methods, it was suggested that NOX2 protein levels was five times higher in csNOX2 TG mice compared to WT mice. Demonstrate the overexpression level of NOX2 protein in csNOX2TG mice hearts.

Author Response

Dear referee number 1.

First, thank you for reviewing our paper and for your thorough and helpful feedback.

In the attached word document we have addressed (in blue) your comments (in black) in the following sections.

Sencerely,

Synne S. Hansen

Reviewer 2 Report

Reviewer Comments (1514988)

In the current manuscript, authors studied the effects of low doses of AngII on hypertension, cardiac function, substrate utilization, and mitochondrial function in both wild type and NOX2 overexpression mice. They found that AngII400 rather than AngII50 led to increased blood pressure, cardiac hypertrophy, and cardiac dysfunction mainly in NOX2 overexpression mice. They concluded that overexpression of NOX2 augments the AngII -induced pathology, with cardiac dysfunction and myocardial metabolic remodeling.

It is a well-written manuscript. However, reviewer does have some concerns about this manuscript.

The major concern: The current manuscript demonstrated the phenomena of AngII treatment, but lack of mechanistic studies.

  • Authors claimed ROS generation contributes to AngII-mediated detrimental effects. However, there were no ROS data in the manuscript.
  • Mitochondrial electron transport chain is the key source of ROS generation. However, authors never studied the mitochondrial ROS generation in wild type and NOX2 overexpression mice.
  • Although authors showed that the rate of OCR was decreased in NOX2 overexpression in basal condition, there were no differences in the rate of OCR between wild type and NOX2 mice when NADH and succinate were used as complex I and complex II substrates, respectively. These data indicated that AngII treatment did not alter mitochondrial function. The results support their effort to study the alteration of enzyme activities in the TCA cycle. Pyruvate dehydrogenase (PDH) is a key enzyme to regulate substrate utilization. Authors already studies the PDK4 gene expression. It is surprised that authors did not measure the activities of PDK4 and PDH in wild type and NOX2 with and without AngII treatment. If AnglI treatment increased PDK4 activity, this will lead to decreased PDH activity. It is hard to understand why AnglI treatment leads to increased glucose oxidation. The quality of manuscript will be greatly improved if authors can show the alteration of PDH in wild type and NOX2 mice with or without AngII treatment.

Minor:

Line 169 and 176. There were no baseline data in Table 1.

Line 222, please specify the comparison between the groups.

Figure 3, it is better to use the ∆ change data rather than show ROX data in different column.

Author Response

Dear referee number 2.

First, thank you for reviewing our paper and for your thorough and helpful feedback.

In the attached word document we have addressed (in blue) your comments (in black) in the following sections. Due to the limited time we have had to make amendments to the manuscript before resubmission, we have not been able to alter the research design of the study. We have however included a limitation section and included amendments to the method, result and discussion section.

Sincerely

Synne S. Hansen

Round 2

Reviewer 2 Report

My concerns have been properly addressed.